# Understanding the Psychological Well-Being of International Arrivals in a Purpose-Designed Australian COVID-19 Quarantine Facility

**DOI:** 10.3390/ijerph192416553

**Published:** 2022-12-09

**Authors:** David Mitchell, Danielle Esler, Kylie Ann Straube, David P. Thomas, Dianne Stephens, Daniel Bressington

**Affiliations:** 1Top End Mental Health Service, Royal Darwin Hospital, Darwin, NT 0810, Australia; 2Public Health Directorate, NT Health, Darwin, NT 0800, Australia; 3Menzies School of Health Research, Charles Darwin University, Darwin, NT 0810, Australia; 4CDU Menzies School of Medicine, Charles Darwin University, Ellengowan Drive, Darwin, NT 0909, Australia; 5College of Nursing and Midwifery, Charles Darwin University, Ellengowan Drive, Darwin, NT 0909, Australia

**Keywords:** COVID-19, quarantine, psychological distress, depression, anxiety, pandemic

## Abstract

Equivocal evidence suggests that mandatory supervised quarantine can negatively affect psychological well-being in some settings. It was unclear if COVID-19 supervised quarantine was associated with psychological distress in Australia. The sociodemographic characteristics associated with distress and the lived experiences of quarantine are also poorly understood. Therefore, this study aimed to evaluate the mental well-being of international arrivals undergoing supervised COVID quarantine in a purpose designed facility in the Northern Territory, Australia. We conducted a concurrent triangulation mixed-methods study comprising of an observational cross-sectional survey (*n* = 117) and individual qualitative interviews (*n* = 26). The results revealed that several factors were associated with distress, including significantly higher levels of depression for those who smoked, drank alcohol, had pre-existing mental health conditions and had no social networks in quarantine. Levels of psychological distress were also related to waiting time for re-entry (the time between applying to repatriate and returning to Australia) and flight origin. Qualitative data showed that despite quarantine being viewed as necessary, unclear communication and a perception of lack of control were affecting emotional well-being. This information is useful to inform the further development of models to identify those at most risk and support psychological well-being in quarantine settings.

## 1. Introduction

Quarantine is the process of separating people who have or may have been exposed to a communicable disease for a duration of time to reduce the risk of spread of the disease to the rest of the population [1]. From 28 March 2020 to February 2022 the Commonwealth Government of Australia retained strict International border controls to prevent the spread of COVID-19 that required people arriving from an international location to quarantine for 14 days [2]. Within Australia’s federated system, states and territories retained control of the public health response to the COVID-19 pandemic. During 2020 this led to a complex, heterogeneous response across the country’s six states and two territories, including varying requirements for quarantine for domestic travellers crossing state and territory borders. In the Northern Territory (NT), international arrivals were quarantined at the Centre for National Resilience (previously known as the Howard Springs Quarantine Facility) in Howard Springs 30 km from the centre of Darwin, the setting for this study. The NT is large and sparsely populated, with a population of 233,000 living in an area slightly larger than South Africa or twice the area of Texas, and more than half living in the capital city Darwin [3].

This study explores the psychological distress of people during quarantine at this facility after arriving from an international location. The study participants were travellers that were part of the Australian government’s repatriation program of Australian citizens and visa eligible family members. The Department of Foreign Affairs and Trade (DFAT) had prioritised their repatriation, among many thousands of Australian citizens stranded overseas as a result of border measures, by a screening process with the intention of selecting the most vulnerable applicants. The protocol for the prioritisation of whom was repatriated has not been made publicly available. Quarantine at the Centre for National Resilience’s part in this national effort to repatriate Australian citizens was as a staging point before movement into the rest of the country.

Concerns about the mental health of individuals in quarantine in Australia was highlighted early in the pandemic. In 2020 the National review of Hotel Quarantine recommended early screening with mental health assessment tools. Ideally immediate concerns were to be promptly identified and with daily follow up established until the psychological issue resolved [2].

Screening for mental distress is crucial because a meta-analysis of individuals experiencing isolation or quarantine demonstrated an increased risk for adverse mental health outcomes, particularly after containment duration of 1 week or longer [4]. However, psychological distress has varied considerably between the recent studies evaluating these models of quarantine, occurring in a range of countries and settings. This has included home, hotels and public spaces [5,6,7,8,9,10]. In an Australian study of quarantine at “medi-hotels” in South Australia in 2020, levels of psychological stress were low [5]. Another study from China found that gender, marital status, previous mental health diagnosis and poor understanding of quarantine policies and processes were associated with increased depression during quarantine [6]. Whereas, distress appeared relatively high within involuntary quarantine facilities in Saudi Arabia, as well as being higher for females and those with pre-existing mental illness [7]. The reported levels of psychological distress were also elevated in state managed facilities in Qatar [8]. The variation in distress levels highlights the heterogeneity of quarantine experienced for COVID-19 mitigation. Therefore, there is a need to better understand the contributing factors influencing psychological distress in these facilities and accurately identify vulnerable cohorts and those at risk. Furthermore, exploring the psychological distress beyond prevalence data and risk factors via qualitative enquiry or a mixed methods approach would afford a deeper understanding of the support needs of quarantined individuals, but such evidence is sparse. A Nepalese quarantine study used qualitative interviews to capture the experience of quarantine [9]. However, the transferability of findings to the Centre for National Resilience is highly questionable. All these gaps led us to ponder what was the psychological well-being for people in quarantine at the Centre for National Resilience, what the experiences that challenge mental well-being were, and how this compared to other studies. Thus, the study aimed to evaluate the psychological well-being and predictors of distress for international arrivals undergoing supervised at this purpose-built facility in the NT, Australia.

### Objectives

The first objective was to quantify the levels of distress in terms of prevalence within people in quarantine and its relationship to potential associated factors. This included sociodemographics and international point of entry into the NT. The intent being to better understand potential determinants as well as identify risk factors.

The second objective was to explore the lived experience of people in supervised quarantine and the impact of quarantine on their mental well-being.

## 2. Materials and Methods

### 2.1. Study Design

We designed a concurrent triangulation mixed methods study comprising of a cross-sectional survey and individual qualitative interviews with people in supervised quarantine. Previous studies of COVID-19 quarantine in Australia report relatively low levels of psychological distress among quarantined people [5]. Hence, we wanted to capture the lived experience of quarantine and how that may affect mental well-being beyond the quantitative data. The qualitative interviews were intended to explore potential sub-clinical distress and influences on mental well-being to enhance our understanding of the survey findings.

### 2.2. Study Setting

Originally, the Centre for National Resilience was developed as a workers’ village in 2012. It serviced 3000 construction staff for the Inpex LNG gas plant. In 2018 it was taken over by the NT Government. In February 2020 it was repurposed as a designated quarantine facility for Australians repatriated from the initial lockdown in Wuhan, China. The Centre for National Resilience is located rurally, approximately 30 km from the centre of Darwin. The accommodation was arranged in groupings of 4 self-contained rooms in blocks across 60 acres. There was a balcony and an ensuite bathroom in each room. Infection control and prevention measures were implemented at all times and all stages of the quarantine facility. Since there were no internal corridors and excellent ventilation within the accommodation, the risk of the virus spreading within the complex was markedly reduced. This has been the cornerstone of the Centre for National Resilience success in mitigating the spread of COVID-19.

Psychosocial supports were available on-site via referral from the nursing staff or self-referral via phone contact. The welfare team consisted of welfare workers, support workers and where required social workers. Specialist mental health services were available by referral from primary health care services, General Practitioners or self-referral via the NT wide Mental Health line. The Centre for National Resilience welfare team was accessible via telephone, video conferencing and where clinically indicated in person, donning Personal Protective Equipment (PPE). Additional to individual services, the welfare team provided general information and service contact details via the Centre for National Resilience information booklet that was provided to all residents at the commencement of their stay. Visiting services from the Royal Darwin Hospitals’ Mental Health Assessment Team (MHAT) included mental health nurses, social workers, and Aboriginal health workers and where clinically appropriate psychiatrists. Visits to the site by the MHAT occurred if deemed clinically necessary via the MHAT service entry criteria. The use of alcohol was prohibited at the facility. Smoking was allowed outside, and smokers were cohorted as much as possible. If a quarantined individual needed medical or psychosocial support related to substance withdrawal this was available via the Centre for National Resilience welfare team and subsequent referral to specialist services, as appropriate.

### 2.3. Participants and Data Collection

Potential participants were eligible for inclusion if they were international arrivals, adults (i.e., over 18 years), able to read and understand English and had capacity to provide informed consent. The survey and interviews were conducted in the first few days of the participants’ second week in quarantine. We calculated that to detect a 25% difference in scores on the 21-item Depression, Anxiety and Stress Scale (DASS-21) across different demographic groups using unpaired T-tests we would require 32 subjects (16 in each group) when assuming 80% power and an alpha value of 0.05. However, given the potential to conduct a regression analysis (dependent on meeting assumptions), we were guided by Harris’s (1985) rule of thumb that an absolute minimum of 10 participants per predictor variable is required for non-linear regression models with six or more predictors [11,12]. Therefore, as 11 potential predictor variables were identified we inflated the minimum sample size requirement to 110 participants. However, given the relatively low levels of psychological distress and low frequency cell counts for psychological distress severity categories that became apparent after data collection we did not conduct regression analysis. In terms of the qualitative interview sample size requirements, we expected to reach data saturation in the qualitative part of the study after 20 participants. Recognising that a-priori determination of qualitative sample sizes is problematic when using an inductive thematic analysis approach [13], we utilised concurrent data collection and analysis to monitor when no new themes were emerging to determine that further interviews were unnecessary.

The research staff visited the study setting regularly, usually weekly. The researchers engaged with the quarantine staff to determine which individuals in quarantine were between days 7–10 of their quarantine period before approaching these potential participants. If they agreed to participate, they were given a survey to complete, which was then collected the same day and kept in a sealed bag until decontaminated. If they also participated in a qualitative interview, it was audio recorded and transcribed and lasted approximately 40 min. The maximum interview duration was limited because they needed to be conducted in full personal protective equipment and outdoors in the tropical heat. The interviews were completed by a research nurse trained in mental health, using techniques to gain rapport, engagement and empathic listening.

A semi-structured interview template was formulated in order to guide the interviews. The template was not prescriptive but served as a guide. This prompted the interviewer to explore the participant’s experience of quarantine and the psychological issues underpinning this. Questions served as a starting point. Many factors were considered in the design of the interview. This included testing for timing and application. Other important considerations were the intended thematic analysis, avoidance of the researcher influencing participants’ responses and allowing the participant to lead the discussion.

Overall, 154 persons in quarantine were approached about the study and 117 completed the survey (75.97% response rate). The surveys were collected between 22/7/21 and 23/12/21. A total of 26 individual qualitative interviews (16 females and 10 males) were completed between the 17/9/21 and the 21/12/21 after 33 people were approached and 7 declined. Interview participants also completed the DASS-21 and their data are included in the survey analyses. Qualitative analysis was conducted concurrently to data collection. Thematic saturation occurred at 12 interviews (7 Germany and 5 UK) and a run of 2 more interviews (2 UK) were conducted to confirm saturation (i.e., that no new themes could be identified).

### 2.4. Ethical Considerations and Approval

Before commencement, the study was approved by the NT Department of Health and Menzies School of Health Human Research Ethics Committee (2020-3761). A research nurse, not otherwise engaged with the facility, explained the study to potential participants, provided a detailed pamphlet, and obtained written consent. Involvement in the study was entirely voluntary and declining to be involved in the project did not impact upon care in the facility.

### 2.5. Study Variables

The DASS-21 was used as a screening tool for psychological distress. It has been well validated as a tool for quantifying levels of depression, anxiety and stress [14,15,16,17]. This is also culturally validated across a range of ethnicities [18]. The tool is simple to administer and can be self-completed. It takes only a few minutes to answer the 21 items and the participant responds on a Likert scale of 0 to 3 (never, sometimes, often and almost always). With 7 items for each dimension of Depression, Anxiety and Stress, scores can range from 0–21 for each dimension. Given the Northern Territory’s large Aboriginal population, the modified PHQ-9 for Aboriginal and Torres Strait Islander people was available as an equivalent screening tool [19]. However, no participants identified as Aboriginal or Torres Strait Islander people.

All participants also completed questions about basic demographic information issues considered relevant to psychological distress in quarantine from previous research and the experience of the authors: age, gender, smoking status, alcohol status, education level, medical condition (yes/no), mental health condition (yes/no), flight origin, networks in quarantine (yes/no), age and time waiting for entry into Australia.

### 2.6. Statistical Analysis

All analyses were performed using IBM SPSS version 26. Non-parametric analyses were used due to significant skewness across all three dimensions (Depression, Anxiety and Stress). Therefore, Independent-Samples Mann-Whitney U Tests were used to test differences in the distribution of total subscale scores between two groups, Independent-Samples Kruskal-Wallis Tests were used to determine differences in the distribution of total subscale scores across multiple groups and Spearman’s Rho correlations were used to test for relationships between the subscale scores and continuous independent variables. The tests were conducted using independent variables previously reported to be associated with differences in psychological distress in quarantined individuals and the general population: age, gender, smoking status, alcohol status, education level, medical condition (yes/no), mental health condition (yes/no), flight origin, networks in quarantine (yes/no), and time waiting for entry into Australia. Significance levels were adjusted using the Bonferroni correction to reduce the risk of type I errors resulting from conducting multiple tests (i.e., by dividing 0.05 by the number of comparisons or correlation tests). Thus, conservative levels of statistical significance (2-tailed) were set at *p* ≤ 0.006 for group comparisons and *p* ≤ 0.025 for correlations.

### 2.7. Qualitative Interview Analysis

Thematic analysis as outlined by Braun and Clarke [20] was used to analyse the interview data. Anonymized interviewees were given a deidentified number. Two researchers familiarized themselves with the data, repeatedly re-reading the transcripts. Both researchers independently generated a list of initial codes from the interview data. Coding was compared. Coding differences were resolved through discussion. Where there was contention, the codes were discussed with the wider research team for resolution. The analysis was inductive reflexive of the researchers’ background knowledge and potential biases. Due to the variation in flight origins and the staggered nature of their arrival (Germany, UK, Canada, Turkey and Singapore) we decided to extend interviews beyond apparent thematic saturation to ensure that we did not miss the experiences of those undergoing quarantine who arrived from different destinations and hence mitigate the risk that new themes might not be discerned.

The triangulation of the qualitative and quantitative data findings was conducted to gain a broad deeper understanding of the phenomena of psychological distress in quarantined individuals (for example, to better understand some of the themes related to influences on the psychological distress reported by participants in the survey), rather than to provide specific explanations for individual interviewees’ distress scores.

## 3. Results

### 3.1. Participant Characteristics

All participants were permanent Australian residents. We collected demographic information relating to gender, education, social networks in quarantine, flight origin, existing mental and physical health conditions, smoking status and alcohol status. Their demographic characteristics are detailed in Table 1.

The participants’ ages ranged from 20 to 88 years (mean = 37.61, SD = 14.39) and on average they had been waiting 15.38 months (SD = 7.35, range = 1–32) for entry into Australia. The majority (60%) were female and just under half (44%) had at least an undergraduate qualification. The largest proportion (a third) flew in from London and the minority had a pre-existing physical (24%) or mental health condition (16%). Just under a third regularly drank alcohol and one in three smoked tobacco. Most had no social networks in quarantine (i.e., no friends or family were in the same quarantine facility).

### 3.2. DASS-21 Scores

The participants’ DASS-21 scores representing their psychological distress were characterized as normal, mild, moderate, severe or extremely severe according to the recommended cut-off scores, as indicated in Table 2.

The participants’ DASS-21 scores are summarized descriptively in Table 3 and the frequency of distress severity is presented in Table 4.

The mean depression, anxiety and stress subscale scores were low overall. The vast majority of participants reported a normal level of psychological distress. Only three participants scored higher than normal for stress and five for depression. Meanwhile, anxiety levels were above normal for 17 participants. 

### 3.3. Relationships between Psychological Distress and Participant Characteristics

The potential relationships between participant characteristics and the depression, anxiety and stress subscale scores were explored using non-parametric analyses. Please see Table 5 for the differences across groups.

The depression total subscale scores were significantly higher for those who smoked, drank alcohol, had pre-existing mental health conditions and with no social networks in quarantine. There were also significant differences in the distribution of total depression score across different categories of education level and flight origin.

The anxiety subscale scores significantly differed across categories of education level and flight origin. The anxiety scores were also found to be significantly higher in those participants with pre-existing mental health conditions.

There were significant differences in the distribution of stress scores across different categories of education level and flight origin. Stress was also significantly higher in those who smoked and had pre-existing mental health conditions.

Significant weak negative correlations were found between waiting time for re-entry (time between applying to repatriate and returning to Australia) and depression (*r_s_* (115) = −0.31, 0.003), anxiety (*r_s_* (115) = −0.218, 0.013) and stress (*r_s_* (115) = −0.220, 0.003). Meanwhile, no significant relationships were found between age in years and depression (*r_s_* (115) = −0.104, 0.262), anxiety (*r_s_* (115) = −0.098, 0.295) or stress (*r_s_* (115) = −0.082, 0.382).

### 3.4. Interviews with Quarantined Persons

Most interview participants reported normal levels of distress on the DASS-21, apart from one with mild depression, 4 with mild anxiety, one with moderate anxiety and one with moderate stress. Their mean age was 38.2 years. The embarkation points included Germany (7), United Kingdom (7), Canada (4), Turkey (4) and Singapore (4).

There appeared to be several themes that spoke to the mental health effects of the pandemic more broadly as a stressor and supervised quarantine having an impact on emotional well-being (see Table 6).

### 3.5. Quarantine Serving a Purpose

One of the more positive themes was that quarantine appeared to serve a greater purpose. Quarantine was occurring to prevent the spread of COVID-19 and was keeping the country safe (see quotes in Table 4). In addition, there was a perceived need to accept the authorities’ decision to quarantine: “*government has made the decision to do it then that’s just what we have to do, no point whinging about it*” (Participant 16, Male Vancouver). In contrast a few others refuted this view: “*I think it is a waste of time, the government should have had something in place by now. For me it just an annoyance*” (Participant 20, Male Singapore).

### 3.6. Communication and Information a Barrier to Well-Being

A more critical theme of quarantine was the dissemination of information within the Quarantine Facility. Staff provided too much information that was impractical to read or were not forthcoming about of areas of concern. When those in quarantine called for assistance, they were told to refer to the manual and orientation. Linked to this were other issues such as staff with English as a second language and feeling it was hard to communicate with them. This appeared to cause psychological unease. Conversely individuals with English as a second language in quarantine, even when fluent, struggled: “*A lot of stress….They did come for the first couple of days to check on me and help but I didn’t have questions then and now I have questions….When I ask for an interpreter they do not get one for me, they just point at the phone number*” (Participant 23, Male Istanbul).

### 3.7. Lack of Agency

One of the broadest themes was lack of control affecting emotional well-being. Once again, this pattern was articulated in multiple domains. This including feeling locked up or in jail, feeling bored, lack of choice of food, a lack of the opportunity to drink alcohol, to move freely and that they were mandated to comply, often many kilometres from their intended destination out of quarantine. Given there was a compulsory financial charge for quarantine that amounted to several thousand Australian dollars there was also a sense of having no control of the cost: “*the cost of this is unreasonable, it has broken me financially, so that is very stressful*” (Participant 8, Female London).

### 3.8. Quarantine an Imposition on Normality

Linked to agency was a feeling that the situation was clearly out the frame of normal life. It was not normal to be stuck in one place for weeks and in relative isolation. It was articulated that it was not normal to be without community and social supports: “*It has been stressful, I have had to leave my wife and kids and go through this without any supports, I am usually a strong person*” (Participant 12, Male London).

### 3.9. Stressful Life Events Precipitating Return to Australia: Quarantine as a Barrier to Resolving Life Events

Many interviewees pointed out that they were not wanting to return from overseas, often intending to return shortly. However, social events had intervened to cause the repatriation such as a sick relative or death of a family member, obligating them be with family to resolve the crisis. Quarantine then became a stress, another barrier to returning home and prolonging the crisis: “*I had to come back cause mum’s sick and needs to have some surgery, I’ve been away the last 4 years*” (Participant 1, Male Frankfurt).

## 4. Discussion

The aims of this study were to better understand the psychological well-being of people undergoing supervised quarantine in a uniquely purposed built facility in the remote region of the NT, Australia. This was accomplished using both quantitative and qualitative data. The study’s mixed method approach has allowed us to better understand both of these objectives.

Psychological distress (albeit at relatively low levels) was found to be associated with the pre-quarantine use of alcohol and tobacco and several socio-demographic characteristics, including education level, flight origin, presence of pre-existing mental health conditions and a lack of social networks in quarantine. Shorter wait times for re-entry were also related to higher levels of distress. The qualitative lived experience of quarantine showed there were themes of quarantine serving a purpose. This may have mitigated distress in some people in quarantine. This was juxtaposed against themes about lack of agency, deviation from normal life and poor communication contributing to distress. In addition, that stressful life events precipitated the repatriation to Australia, preloading some participants for psychological distress.

In this study, the overall mean levels of distress as measured DASS-21 appeared low when compared to other recently studied COVID-19 quarantine facilities [6,7,8,9,10]. Comparisons of these findings with previous literature is, however, complicated by the heterogeneous nature of quarantine and study designs. There have been many approaches to quarantine of international arrivals during the pandemic and different types of quarantine facilities. It has also been adopted in a range of countries and settings. It is, however, reassuring to observe that the Centre for National Resilience appeared comparatively less impactful on psychological distress. Whilst we can only speculate, the living arrangements in this facility may have helped. The accommodations were arranged in single-storied groupings of 4. The rooms were self-contained with a balcony and ensuite bathroom in each room. Those in quarantine appeared to have the opportunity to socialize with others in adjacent blocks, exercise outdoors as well as retreat to the privacy of their room. Whilst such arrangements were designed for better infection control, there may been incidental advantages to psychological well-being. These design features could be considered in future models of quarantine.

Despite the Centre for National Resilience having relatively low levels of psychological distress when compared to studies conducted in other countries, many of the markers that we assessed were significantly higher for subscales of depression, anxiety and stress, indicating a link to increased psychological distress in those individuals in the quarantine facility. In addition to the observed associations of psychological distress with alcohol and smoking, the depression and anxiety subscale scores were significantly higher in those with no social networks in quarantine and in those participants with pre-existing mental health conditions. Stress was also significantly higher in those with pre-existing mental health conditions. The facility could therefore implement strategies to target those who smoke, consume alcohol, and have pre-existing mental health issues for closer scrutiny. Similarly, those who are quarantining alone without the social networks of family, partners or friends in the facility need to be identified for attention and the facility could perhaps establish a peer-support network using video conferencing facilities to provide a social network for people undergoing quarantine at the same time.

An area for further evaluation is the link between psychological distresses and point of flight origin and education level. Both were significantly different between sub-groups and levels of depression, anxiety and stress. For example, there were significant differences in the distribution of total depression score across different categories of education level (*p* = 0.001) and flight origin (*p* < 0.0001). What accounts for this is harder to understand. It is possible that the embarkation point gives rise to a stress “preload”–different departure points were associated with different levels of pre-departure stresses. For instance, the higher distress levels in those embarking from Frankfurt, compared to the other cohorts, may relate to any number of social, political and contextual issues in that region at the time of departure. The association between higher education and distress is an observation that cannot easily be accounted for. Regardless, flight origin and education level could be used to inform detection of those most at risk of psychological distress in future quarantine situations.

The semi-structured interviews improved our qualitative understanding of the stresses associated with supervised quarantine in this facility and how it impacted emotional well-being beyond the prevalence data. There were themes of lack of agency and control as well as themes that quarantine was a breach of normality. Both were perceived as influencing distress. It is arguable that these phenomena may lead to psychological distress. The counter approach would be to improve a sense of agency and normality in quarantined individuals. Providing choice of foods, the ability to access entertainment as well as information technology that encouraged connection with social supports, may have helped. Similarly, the theme of poor communication and information as a barrier could be addressed by improving protocols that prioritized enhanced communication and the dissemination of information. The qualitative data reinforce that promoting enhanced communication may be particularly important for quarantined individuals who do not speak English as a first language, despite them being permanent Australian residents and hence having fluency in English.

It is possible to postulate a link between the qualitative thematic analysis and the quantitative DASS-21 screening results. The depression total subscale scores were significantly higher for those who smoked (*p* = 0.003) and drank alcohol (*p* = 0.006). Stress was also significantly higher in those who smoked (*p* = 0.004). It can be speculated that lack of control and agency may have been a contributing factor. Since those individuals in quarantine were unable to consume alcohol or readily smoke, as part of the facility’s strict regulations, any sense of agency or normal practice was arguably removed. This may have led to psychological distress. In such a scenario, it would be reasonable for the Centre for National Resilience to consider its alcohol consumption and smoking policies. There is an obvious need to balance the emotional well-being of those in quarantine with good public health policy. Responsible alcohol consumption and harm minimization as well as reducing the risks of second-hand smoke exposure need to be incorporated when designing quarantine regulations. In contrast, it is also worth reflecting that drinking and alcohol consumption have often been linked to mental health distress [21,22,23]. Hence, this may be a confounding variable. There is insufficient evidence within this study and the existing literature to definitively answer this.

One of the issues that was revealed from the thematic analysis was that many of the repatriated Australians undergoing quarantine had carried out due to an urgent social crisis. Some individuals were returning due to a range of stressful life events. For example, burying a family member, placing elderly parents in residential care or dealing with significant family financial issues. They were generally distressed by these circumstances and were returning predominantly to deal with these events. There is extensive literature to suggest negative life events influence psychological distress [24,25]. In addition, social determinants influence psychological resilience and well-being in a disaster [26]. In these scenarios, the individuals undergoing quarantine were arguably pre-selected for psychological distress as their individual cases appeared to prioritise them for return. This seemed to align with the quantitative DASS-21 screening results. There were significant weak negative correlations found between waiting time for re-entry and depression (*p* = 0.003), anxiety (*p* = 0.018) and stress (*p* = 0.017). In short, those that appeared to be prioritised for return to Australia and their travel expedited appeared to have higher levels of psychological distress. Thus, shorter waiting times to return were associated with poorer emotional wellbeing. A practical option for the Centre for National Resilience would be to screen waiting times (and so prioritisation of repatriation) as a potential predictor of emotional well-being in quarantine.

Even as the control of COVID-19 has transited away from supervised quarantine, quarantine remains an important potential strategy in the event of future biosecurity threats. A better understanding of the psychological risks is key to ensuring that the mental well-being and psychological safety of those undergoing quarantine are supported.

## 5. Strengths and Limitations

The mixed methodology allowed us to gain a more in depth understanding of the phenomenon. The study was conducted in a real-world setting despite the practical challenges this involved, and the continued recruitment and concurrent qualitative data analysis occurred in real-time to ensure data saturation was met. However, the descriptive nature of the study limits our ability to draw more certain conclusions.

There are several limitations due to methodological issues and the practical challenges of conducting this research in a dynamic emergency setting. For instance, our decision to carry out DASS-21 screening at one point in time, instead of two, was influenced by our inability to access the facility more than once weekly. The convenience sampling of those screened and those completing semi-structured interviews was also influenced by necessity, but negatively impacts the generalisability of the findings. Finally, we did not collect data on participants’ first spoken language, which precluded analysis of a potential relationship between psychological distress and language difficulties.

## 6. Conclusions

This mixed method study informs our understanding of the phenomenon of psychological distress in international arrivals undergoing supervised quarantine in a purpose designed facility for the mitigation of COVID-19 transmission in the NT, Australia. There are a number of psychological stressors that appear to be inherent to the quarantine facility as well as stressors that exist separate to the experience in quarantine. Several risk factors were linked to significantly higher rates of distress as determined by the DASS-21 subscales of depression, anxiety and stress. This included smoking, alcohol use, mental health history and a lack of social networks whilst undergoing quarantine. Those that experienced shorter wait periods before repatriation to Australia were significantly correlated to more distress. It is likely their individual circumstances may have both prioritised them for return, but also pre-selected them as individuals as more likely to be distressed. This information is vital to understanding the psychological factors within the quarantine environment and point to how we further develop the models to support psychological well-being in quarantine settings.

## Figures and Tables

**Table 1 ijerph-19-16553-t001:** The participants’ characteristics.

Demographic	N (%)
Gender	
Male	46 (39.3)
Female	71 (60.7)
Education level	
High school	44 (37.6)
Undergraduate	29 (24.7)
Postgraduate	23 (19.6)
Nil/primary	21 (17.9)
Social networks in quarantine	
Yes	46 (39.3)
No	71 (60.7)
Flight origin	
Frankfurt	17 (14.5)
London	39 (33.3)
Vancouver	23 (19.6)
Istanbul	17 (14.5)
Singapore	21 (17.9)
Smoking status	
Smoker	37 (31.6)
Non/ex-smoker	80 (68.4)
Alcohol status	
Drinker	55 (47.0)
No/ex-drinker	62 (53.0)
Pre-existing mental health conditions	
Yes	19 (16.2)
No	98 (83.8)
Pre-existing physical health conditions	
Yes	29 (24.8)
No	88 (75.2)

**Table 2 ijerph-19-16553-t002:** DASS21 Ratings for Depression, Anxiety and Stress.

Level	Depression	Anxiety	Stress
Normal	0–4	0–3	0–7
Mild	5–6	4–5	8–9
Moderate	7–10	6–7	10–12
Severe	11–13	8–9	13–16
Extremely Severe	14+	10+	17+

**Table 3 ijerph-19-16553-t003:** DASS Descriptive statistics.

DASS Subscale Score	*N*	Mean	Std. Deviation	Range	Mean 95% Confidence Interval		Median
					Lower	Upper	
Depression	117	2.20	2.92	0.00–14.00	1.66	2.73	1.00
Anxiety	117	3.21	4.22	0.00–18.00	2.44	3.99	1.00
Stress	117	5.50	4.74	0.00–19.00	4.63	6.36	4.00

**Table 4 ijerph-19-16553-t004:** The frequency of distress severity.

DASS Subscale	*N*	Normal (Frequency, %)	Mild (Frequency, %)	Moderate (Frequency, %)	Severe (Frequency, %)	Extremely Severe (Frequency, %)
Depression	117	112 (95.7)	4 (3.4)	1 (0.9)	0	0
Anxiety	117	100 (85.5)	5 (4.3)	8 (6.8)	4 (3.4)	0
Stress	117	114 (97.4)	1 (0.9)	2 (1.7)	0	0

**Table 5 ijerph-19-16553-t005:** The differences in psychological distress across groups.

Group	DepressionMean Rank	Z, *p*-Value	Anxiety Mean Rank	Z, *p*-Value	Stress Mean Rank	Z, *p*-Value
Smoking status (N)						
Smoker (37)	72.30	−2.96, 0.003 *	67.74	−1.93, 0.053	72.20	−2.88, 0.004 *
Non/ex-smoker (80)	52.85		54.96		52.89	
Alcohol status						
Drinker (55)	67.85	−2.73, 0.006 *	67.70	−2.67, 0.008	67.73	−2.63, 0.008
No/ex-drinker (62)	51.15		51.28		51.26	
Social networks in quarantine						
Yes (46)	38.13	−5.50, <0.001 *	41.85	−4.50, <0.001 *	40.76	−4.50, <0.001 *
No (71)	72.52		70.11		70.82	
Pre-existing mental health conditions						
Yes (19)	86.87	−4.02, <0.001 *	80.95	−3.15, 0.002 *	80.13	−2.979, 0.003 *
No (98)	53.60		54.74		54.90	
Pre-existing physical health conditions						
Yes (29)	58.03	−0.18, 0.856	60.47	−0.27, 0.784	59.95	−0.174, 0.862
No (88)	59.32		58.52		58.69	
Gender						
Male (46)	52.09	−1.82, 0.069	55.58	−0.90, 0.369	55.55	−0.89, 0.374
Female (71)	63.48		61.22		61.23	
		**Kruskal-Wallis H, *p*-value**		**Kruskal-Wallis H, *p*-value**		**Kruskal-Wallis H, *p*-value**
Flight origin						
Frankfurt (17)	95.00		88.09		81.24	
London (39)	66.78		61.46		68.09	
Vancouver (23)	51.98		61.28		56.93	
Istanbul (17)	44.76		45.03		44.88	
Singapore (21)	34.62	37.95, <0.001 *	39.69	23.51, <0.001 *	37.81	21.51, <0.001 *
Education level						
High school (44)	73.66		75.35		74.98	
Undergraduate (29)	56.64		49.43		54.47	
Postgraduate (23)	47.93		51.09		49.20	
Nil/primary (21)	43.67	15.90, 0.001 *	46.62	17.32, 0.001 *	42.52	17.30, 0.001 *

* *p* ≤ 0.006.

**Table 6 ijerph-19-16553-t006:** The themes from interviews of participants.

Themes	Quotes Demonstrating Themes
Quarantine serving a purpose	“I think it’s good, I know a lot of people complain about (it), but it’s really keeping Australia safe. Most Australians are really naive as to how full on the virus is”. (Participant 11, Female London) “I understand the need for it, I understand the government is doing what they think it right” (Participant 1, Male Frankfurt)
Communication and Information a barrier to well-being	“Well it’s all horrible, they give you heaps of information on-line and in paper form but when you ask for information or help with anything they just tell you to call the help numbers. It’s really frustrating”. (Participant 6, Female Frankfurt) “They just refer you to the orientation booklet or say call this number or email the well-being team. That’s fine if your confident enough to do that, but if you’re not, it’s a fend for yourself situation” (Participant 10, Female London) “whenever you ask the nurses for anything they just tell you to read the manual or call the wellbeing number, it’s not very personal” (Participant 15 Female Vancouver)
Lack of Agency	“It feels like Jail, the only time that is bearable is at night, that’s when I sit on the Veranda” (Participant 1, Male Frankfurt) “The thing that has had a huge impact on my mental health has been food. What is the point of asking us food preference we have if they are not going to be followed?” (Participant 4 Male Frankfurt) “I would kill for a beer and to eat the food I would usually eat” (Participant 5, Male Frankfurt)
Quarantine an imposition on normality	“not sleeping in my own bed or having my usual things around is hard. It’s not my space how I like it and I don’t feel settled”. (Participant 2, Female Frankfurt) “Just being bored and not being about to do the things I usually like to do is the main stress” (Participant 19, Singapore)
Stressful life events precipitating return to Australia: Quarantine as a barrier to resolving life events	“I have a lot of things I needs to get done once I get back to Brisbane so I am a bit stressed about that, but there is nothing really in here that is giving me stress, its more just the anticipation of what needs to be done when I get home, I kind of feel like sitting around here is wasting time when I could be getting important jobs done, but there is nothing I can do about it” (Participant 18, Female Vancouver) “Not only am I having to bury my sister but I am going to have to sort a nursing home for mum. My sister was mums carer so this (travel and quarantine) all happened too quickly” (Participant 12, Male London) “I am trying to get home because my dad has been diagnosed with bowel cancer and my family is basically losing it” and “I think it (quarantine) is a waste of time to be honest, look at all the cases in Victoria and New South Wales” (Participant 4, Female Frankfurt) “It was very stressful to get here and there was a lot involved in getting back to Australia”(Participant 25, Male Istanbul)

## Data Availability

The data is not publicly available due this not been granted by MENZIES HREC as part of the approval. Study participants have not approved sharing the data beyond the study team, so data is not publicly available.

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
