# Peer review of "Understanding the Psychological Well-Being of International Arrivals in a Purpose-Designed Australian COVID-19 Quarantine Facility"

_ijerph, 2022, doi:10.3390/ijerph192416553_

Round 1

Reviewer 1 Report

Dear Authors, please, find my review below:

REVIEW: Understanding the psychological wellbeing of international arrivals in a purpose-designed Australian COVID-19 quarantine facility

I see the issue addressed in this paper as having a growing importance in the field of public health. I hope that the authors will be able to revise it to meet the publication criteria of IJERPH. For the time being, the paper is more like an evaluation statement than a research article.

The paper has many strengths, for example the wider Australian context of the study design is thoroughly enough described in Introduction and the study design and the study setting in Materials and Methods. The tables are easy to understand and so is the appendix about interview themes.

I will concentrate on the ways to improve the manuscript.

My main advice for the authors is to carry on their aim, mentioned in the title, to understand the psychological wellbeing of their participants, throughout the paper.

Introduction should give a proper background to the objectives of the research, based on earlier research. The authors have chosen to present the earlier research almost totally in Discussion, as new perspectives. However, all the references/earlier research results/concept that can been used to formulate the objectives should be mentioned in Introduction.

Another important revision step  is to formulate the research objectives from this very vantage point of understanding the phenomena under study, not from the vantage point of research methods. Research methods should be chosen on the basis of determined what is the best method to be used in answering the research questions.

Materials and Methods -section needs revision in subsections Subjects, Survey data collection and recruitment and Qualitative interviews. The contents are overlapping and intertwined, which calls for one combined section titled for example Participants and data collection, or alternatively, if separate titles are preferred, overlapping should be omitted. It is difficult to see why the interviewees would need a subsection of their own in this matter.  Also, paragraphs describing the number of respondents (Survey) and interviewees (Interviews with Quarantined persons) should be removed from Results -section to the above mentioned new combined subsection. In subsection Qualitative interview analysis it should be mentioned that the anonymized interviewees were given numerical codes.

Results -section would be more easily read, if it would have the following structure: Subtitle a short mention about what results are to be presented in Table X, then the actual Table X and after it the descriptive summaries and first explanations. This can be applied both to the survey and the interview results, but the interview results should already have some hints towards the understanding of the phenomena studied, not only lists of computed data.

Discussion -section could, for example, start with a summary of the key findings and proceed toward interpretations: explanation of the meaning and importance of the findings. The present discussion relates to similar studies and makes comparisons, but the aim of understanding the results is given up. Especially the interview results should be discussed more thoroughly, towards understanding.  Also, some concrete practical implications should be mentioned since the quarantine conditions are very clearly described in the current Introduction. In the unfortunate case of new quarantines, what should be done in different ways and how?

Lines 180 and 189, both include the subtitle Qualitative interviews

Line 248, there should be Table 6, instead of Table 4

All the best for your revisions, Your reviewer

Reviewer 2 Report

Ethnicity is an element that influences the manifestation of emotional alterations during the pandemic, if it is considered that it corresponds to a language that hinders the interactions between the health personnel and the confined persons.

In the quantitative section, the data are analyzed using non-parametric statistics: it can be considered that the finger rule suggested by the authors to establish the sample size (Harris, 1985), is only applied because we are dealing with predictor variables and not with linear regression models.

The criteria of bioethics or ethics in research are respected.

It is possible to expand on how the coding problems were solved.

Take care with the formatting of the tables presented

Use the abbreviation Rho for the correlation between the results of stress and pre-existing conditions of vulnerability.

The authors address several issues in the discussion that are not sufficiently covered in the results (e.g., where participants are from and the language they use). 

The method of triangulation is not entirely clear and defined, the participants who participated in the interviews did not have a characterization of the stress elements that were measured quantitatively, so linking the two results (coming, it seems, from different sources) is not done.

Reviewer 3 Report

I found this an unusually well-written and described study. I have minor comments only where points are not entirely clear:

Please could the authors expand on what they mean by 'waiting time for re-entry'? The process of entry to Australia during the height of the pandemic may not be well-known globally.

I would advise against using unfamiliar acronyms. Writing Centre for National Resilience is better than CNR. DASS-21 is OK but should be written in full the first time it appears (page 3)

The table formatting made it harder to 'see at a glance' the points made about flight origin/education status but this is likely to be resolved at print stage!

The finding that lack of control affected mental wellbeing is not a surprise and is a helpful reminder that enforced behaviour is likely to result in dissatisfaction that the CNR might have reasonably foreseen. Some suggestion of practical changes that could be made in a future scenario would be useful.

The explanation of flight origins and potential reasons for higher distress make sense but the discussion as a whole would be stronger if the authors could make reference to psychological literature that supports their arguments,particularly those regarding 'preloading'/other social factors affecting mental state. 

There are a couple of sentences that don't make sense. Authors to check:

* page 10, line 294

* page 11, line 282-4 

Round 2

Reviewer 1 Report

Dear Authors,

I was  happy to notice that you did your revision work conscientiously. As I see it, your paper has now a logical plot and is easy to comprehend.

I will recommend the publication of your paper to the editors.

Your reviewer